# Exogenous Melatonin Application Accelerated the Healing Process of Oriental Melon Grafted onto Squash by Promoting Lignin Accumulation

**DOI:** 10.3390/ijms25073690

**Published:** 2024-03-26

**Authors:** Yulei Zhu, Jieying Guo, Fang Wu, Hanqi Yu, Jiahuan Min, Yingtong Zhao, Changhua Tan, Yuanwei Liu, Chuanqiang Xu

**Affiliations:** 1College of Horticulture, Shenyang Agricultural University, Shenyang 110866, China; zyl@stu.syau.edu.cn (Y.Z.); jieying336@stu.syau.edu.cn (J.G.); wufang366@stu.syau.edu.cn (F.W.); hansyu@stu.syau.edu.cn (H.Y.); mjh@stu.syau.edu.cn (J.M.); zyt@stu.syau.edu.cn (Y.Z.); tanchanghua@syau.edu.cn (C.T.); 2Key Laboratory of Protected Horticulture (Ministry of Education), Shenyang Agricultural University, Shenyang 110866, China; 3Modern Protected Horticultural Engineering & Technology Center, Shenyang Agricultural University, Shenyang 110866, China; 4Key Laboratory of Horticultural Equipment (Ministry of Agriculture and Rural Affairs), Shenyang Agricultural University, Shenyang 110866, China; 5College of Horticulture, Hunan Agricultural University, Changsha 410128, China; liu_yyww@163.com

**Keywords:** melatonin, melon, squash, graft union healing, lignin, *CmCAD1*

## Abstract

Melatonin (MT) is a vital hormone factor in plant growth and development, yet its potential to influence the graft union healing process has not been reported. In this study, we examined the effects of MT on the healing of oriental melon scion grafted onto squash rootstock. The studies indicate that the exogenous MT treatment promotes the lignin content of oriental melon and squash stems by increasing the enzyme activities of hydroxycinnamoyl CoA ligase (HCT), hydroxy cinnamaldehyde dehydrogenase (HCALDH), caffeic acid/5-hydroxy-conifer aldehyde O-methyltransferase (COMT), caffeoyl-CoA O-methyltransferase (CCoAOMT), phenylalanine ammonia-lyase (PAL), 4-hydroxycinnamate CoA ligase (4CL), and cinnamyl alcohol dehydrogenase (CAD). Using the oriental melon and squash treated with the exogenous MT to graft, the connection of oriental melon scion and squash rootstock was more efficient and faster due to higher expression of wound-induced dedifferentiation 1 (*WIND1*), cyclin-dependent kinase (*CDKB1;2*), target of monopteros 6 (*TMO6*), and vascular-related NAC-domain 7 (*VND7*). Further research found that the exogenous MT increased the lignin content of the oriental melon scion stem by regulating *CmCAD1* expression, and then accelerated the graft healing process. In addition, the root growth of grafted seedlings treated with the exogenous MT was more vigorous.

## 1. Introduction

Grafting technology can potentially enhance the resistance of plants against both biotic and abiotic stresses, mitigate soil-borne diseases, and overcome the challenges of continuous cropping. This technique has been extensively utilized in cultivating fruits and vegetables [1,2]. The oriental melon is a variety of melon with rich nutritional value and excellent taste. However, fusarium wilt disease occurs severely in production. Grafting is one of the most effective methods to prevent fusarium wilt disease in greenhouse melon production. Currently, grafting cultivation of melons has been widely adopted in production. It is widely recognized that the establishment of successful healing of rootstocks and scions is a crucial prerequisite for the cultivation of grafted seedlings. The healing process in grafted plants is a complex physiological mechanism that involves formation of isolation layers, callus generation, and vascular bundle reconnection [3]. According to Tan et al. [4], melatonin (MT) is currently recognized for its potent antioxidant function as an endogenous scavenger of free radicals. Meanwhile, it also plays a significant role in the growth and development of plants by enhancing seed germination, promoting root growth, and regulating callus formation [5,6,7]. Exogenous MT could effectively improve the ability of plants to resist biotic and abiotic stresses. Studies have demonstrated that the application of exogenous MT could amplify the resistance and drought resistance of apple plants to apple fruit spot disease and enhance the heat resistance of *Arabidopsis* and the low-temperature resistance of *Rhodiola* and *Ulmus pumila* [8,9,10,11,12,13]. However, it remains unknown whether exogenous MT application can regulate the graft union healing process of the plants.

Recent studies have reported that exogenous MT could significantly promote plant lignin accumulation [14,15,16]. Lignin played an essential role in plant growth and development and was actively involved in plant responses to various environmental stresses in [17,18,19,20]. The synthesis of lignin was significantly regulated by the enzyme activity of cinnamyl alcohol dehydrogenase (CAD), which was found to be an essential enzyme in the lignin synthesis pathway. However, it is even unclear whether exogenous MT treatment might improve lignin accumulation by inducing the up-regulated expression of *CmCADs* in melon. Furthermore, transcriptome analysis on grafted litchi chinensis and grafted melon showed that genes related to lignin biosynthesis were differentially expressed during the graft union healing process [21,22]. We speculated that lignin might be involved in the healing of graft unions, but further studies were required to back this up. Specifically expressed genes at the graft union of plants participated in a series of physiological and biochemical processes during graft healing, such as some genes involved in secondary cell metabolism, cell wall synthesis, vascular tissue reconnection, and other functions [23]. Recently, many key factors have been identified as regulating the graft union healing of plants. The genes of WIND1-4 were crucial in regulating wound-induced callus formation by promoting cellular dedifferentiation and proliferation [24,25]. The genes of WOX played an essential role in many physiological processes, including root and stem apical meristem formation, organ development, and vascular tissue development [26,27,28,29]. *SlWOX4* potentially regulated the compatibility of scion and rootstock, which was crucial for vascular bundle reconnection in the graft union healing of plants [30]. TMO6 was closely related to callus formation and vascular regeneration of grafted unions during the healing process [31], and VND7 directly or indirectly induced the expression of some genes associated with xylem vessel element differentiation [32,33]. However, it has not been reported whether exogenous MT application could affect graft union healing by regulating the expression of these genes related to graft union healing.

In this study, we applied the exogenous MT to treat the oriental melon scion and squash rootstock seedlings and investigated the effects of the exogenous MT on lignin accumulation and graft union healing process. Furthermore, we hope to provide a theoretical and practical basis for regulating graft union healing by the exogenous MT.

## 2. Results

### 2.1. Identification of Members of the CmGH9B Gene Family Members

We found that exogenous MT treatment could significantly increase the lignin contents of oriental melon and squash seedling stems (Figure 1A). The enzyme activities related to lignin biosynthesis, like HCT, HCALDH, COMT, CCoAOMT, 4CL, and CAD, were determined. The results showed that the HCT (Figure 1B), HCALDH (Figure 1C), COMT (Figure 1D), CCoAOMT (Figure 1E), and CAD activities (Figure 1H) of the oriental melon and squash seedling stems were increased by the exogenous MT treatment. The PAL activities (Figure 1F) of the oriental melon seedling stems were increased by exogenous MT treatment. However, exogenous MT did not significantly affect the 4CL activities of the oriental melon scion and squash seedling stems (Figure 1G). The exogenous MT applications could promote lignin accumulation in stems of oriental melon and squash seedlings by increasing enzyme activities related to lignin biosynthesis. 

### 2.2. Historic Observation of the Graft Union Healing Process of the Oriental Melon Scion Grafted onto Squash Rootstock by the Exogenous MT Treatment

In the acid fuchsin absorption experiment, the oriental melon scion and squash rootstock seedlings treated with the exogenous MT were used to graft. As the oriental melon scion and squash rootstock gradually heal, acid fuchsin will move along the vascular bundle and eventually appear red in the stem tissue sections of the oriental melon scion. At 8 DAG, we distinctly observed the acid fuchsin in the stem tissue sections of the exogenous MT treatment. However, it appeared in the stem tissue sections of the control at 9 DAG (Figure 2). To further investigate the effect of exogenous MT treatment on graft healing, we conducted the paraffin section test. Under the exogenous MT treatment, the callus formation happened at 5 DAG, and the vascular bundles connected at 8 DAG. However, the callus formed at 6 DAG and the vascular bundles joined at 9 DAG in the control (Figure 3). So, the exogenous MT treatment not only promoted the lignin content of oriental melon and squash seedlings but also accelerated the graft union healing process.

### 2.3. Expression Profiles of the Genes Related to Graft Union Healing of the Oriental Melon Scion Grafted onto Squash Rootstock by the Exogenous MT Treatment

To further analyze the effects of exogenous MT treatment on the graft union healing process, we determined the expression of the genes related to healing, including WOUND-INDUCED DEDIFFERENTIATION1 (*WIND1*), WUSCHEL-RELATED HOMEOBOX4 (*WOX4*), CYCLIN-DEPENDENT KINASE (*CDKB1;2*), TARGET of MONOPTEROS 6 (*TMO6*), and VASCULAR-RELATED NAC-DOMAIN 7 (*VND7*) of the oriental melon and squash rootstock (Figure 4 and Figure 5). The relative expression levels of *CmWIND1* of oriental melon (Figure 4A) and *CmoWIND1* of squash rootstock (Figure 5A) increased gradually from NG to 2 DAG. The relative expression levels of *CmWIND1* in the oriental melon scion treated with exogenous MT were significantly higher than the control at 2 DAG. The relative expression levels of *CmoWIND1* in the squash rootstock treated with exogenous MT were also significantly higher than the control at 1 and 2 DAG. There was no significant difference in the relative expression levels of *CmWOX4* between MT and control, even though they increased from 1 to 3 DAG (Figure 4B). And the relative expression levels of *CmoWOX4* in the squash rootstock treated with exogenous MT were significantly higher than the control at 1 DAG (Figure 5B). From 4 to 6 DAG, the relative expression levels of *CmCDKBB1;2* showed a decreasing trend (Figure 4C), and the relative expression levels of *CmoCDKB1;2* showed an increasing trend (Figure 5C). 

However, the exogenous MT treatment significantly improved the relative expression levels of *CmCDKB1;2* and *CmoCDKB1;2* at 6 DAG. From 6 to 8 DAG, the relative expression levels of *CmTMO6* first increased and then decreased. Those of the oriental melon scion treated with exogenous MT were significantly higher than the control at 6 and 7 DAG (Figure 4D). The relative expression levels of *CmoTMO6* changed slightly, and the exogenous MT significantly improved the relative expression levels of *CmoTMO6* at 6 DAG (Figure 5D). From 8 to 10 DAG, the relative expression levels of *CmVND7* in oriental melon scion treated with the exogenous MT were maintained at high levels. They were significantly higher than the control at 8 and 9 DAG (Figure 4E). Then, the relative expression levels of *CmoVND7* in the squash rootstock treated with the exogenous MT were significantly higher than the control at 8 DAG (Figure 5E).

### 2.4. Expression Profiles and Functions of CmCADs during the Graft Healing Process of Oriental Melon Scion Grafted onto Squash Rootstock by the Exogenous MT Treatment

CAD is the rate-limiting enzyme in lignin biosynthesis. We identified the *CmCAD1*, *CmCAD2*, *CmCAD3*, and *CmCAD4* from the melon genome. We detected their expression profiles from 2 to 9 DAG (Figure 6). 

At 2 DAG, the relative expression levels of *CmCAD1*, *CmCAD2*, *CmCAD3,* and *CmCAD4* by the exogenous MT treatment were significantly higher than the control. Moreover, their relative expression levels significantly declined at 5 DAG. From 5 to 9 DAG, the relative expression levels of *CmCAD1* gradually increased. The exogenous MT treatment significantly improved *CmCAD1* expression, except for 5 DAG (Figure 6A). The relative expression levels of *CmCAD2* by the exogenous MT treatment were markedly higher than the control at 5 and 6 DAG (Figure 6B). The relative expression levels of *CmCAD3* and *CmCAD4* by the exogenous MT treatment were significantly higher than the control at 6 DAG and 8 DAG, respectively (Figure 6C,D). However, their relative expression levels were markedly lower than the control at 9 DAG. According to the results, we concluded that *CmCAD1* was a crucial gene in lignin biosynthesis and involved in the graft union healing process.

To further verify the function of *CmCAD1* during the graft union healing of oriental melon scion grafted onto squash rootstock, we performed a transient silencing assay of *CmCAD1* in the oriental melon scion stems. The results indicated that the *CmCAD1* expression levels in the transiently silenced oriental melon stem were significantly lower than the control during the graft healing process (Figure 7B), and the lignin content of those was also markedly lower than the control (Figure 7A). In addition, we found that the acid fuchsin was observed in the at 9 DAG and in the control at 8 DAG. Furthermore, we found that there were more acid fuchsins in the control than in the TRV-*CmCAD1* at 10 DAG (Figure 8). Obviously, the vascular connectivity of the TRV-*CmCAD1* was weaker than the control. In conclusion, *CmCAD1* might affect the graft union healing of oriental melon scion grafted onto squash rootstock by regulating lignin biosynthesis.

### 2.5. Effects of the Exogenous MT Treatment on the Grafted Seedling Root Growth of Oriental Melon Scion Grafted onto Squash Rootstock

The quality of the grafted seedlings is a key factor in achieving a high quality and yield of oriental melon. In this study, we also analyzed the root growth characteristics of five-leaf grafted seedlings treated with exogenous MT using the root system scanner (Figure 9A). The results indicated that the root length (Figure 9B), root surface area (Figure 9C), root average diameter (Figure 9D), the total number of root tips (Figure 9E), and root forks of grafted seedlings treated with exogenous MT (Figure 9F) were all significantly higher than those of the control. The roots of the exogenous MT-treated seedlings grew better than those of the control.

## 3. Discussion

The phenylpropane metabolic pathway was the most important metabolic pathway in plant resistance to biotic and abiotic stresses, and lignin biosynthesis was a downstream branch of the phenylpropane metabolic pathway [34,35]. Studies have shown that exogenous MT is involved in lignin biosynthesis. Li et al. [36] reported that the exogenous MT enhanced the resistance of cotton to verticillium wilt by up-regulating the expression of lignin and gossypol synthesis-related genes in the phenylpropane metabolic pathway, mevalonic acid pathway, and gossypol synthesis pathway. Qu et al. [37] found that the lignin contents of blueberry fruit were promoted by increasing the activities of PAL, C4H, 4CL, CAD, PPO, and POD after soaking in the exogenous MT solution. In this study, we measured the lignin contents of oriental melon scion and squash rootstock after the exogenous MT treatment and found that the lignin contents were significantly increased (Figure 1A) by increasing the related enzyme (HTC, HCALDH, COMT, CCoAOMT, PAL, CAD) activities (Figure 1B–H). The results were similar to those of Li et al. [36] and Qu et al. [37]. However, it remains unclear whether the increase in lignin content in the oriental melon scion stem will promote the graft union healing of oriental melon grafted onto squash rootstock. Some reports indicated that the success of plant grafting depended on the effective connection of the vascular bundle between the scion and rootstock [38,39]. We found that the exogenous MT treatment accelerated the graft union healing process through the acid fuchsin absorption and paraffin section tests (Figure 2 and Figure 3). This suggested that exogenous MT treatment is beneficial for the graft union healing of oriental melon grafted onto squash rootstock. Cinnamyl alcohol dehydrogenase (CAD) is a vital enzyme function at the last step in the lignin synthesis, and jasmonic acid (JA) increased the expression levels of *CmCADs* to regulate the lignin deposition in the melon stems [40]. In this study, the exogenous MT treatment could significantly increase the *CmCAD1* expression (Figure 6A), and the graft union healing process was delayed after the *CmCAD1* silencing in the oriental melon stems (Figure 7 and Figure 8). The results indicated that the positive influence of exogenous MT-induced lignin deposition was advantageous for the graft union healing process of oriental melon scion grafted onto squash rootstock. Nevertheless, further investigation requires a comprehensive understanding of the internal regulatory mechanisms governed by exogenous MT treatment. Exploring these intricacies will contribute to a more nuanced comprehension of how exogenous MT actively participated in and regulated the graft union healing, paving the way for potential applications in optimizing and enhancing this essential process in plants.

The plant hormones IAA, CTK, and GAs, were closely related to callus formation, vascular bundle development, and reconnection during the graft union healing process [41,42,43]. Some reports showed that applying exogenous naphthylacetic acid (NAA) could significantly promote graft union formation [44]. It has been reported that several essential genes play a role in regulating graft union healing. WIND1 regulates the callus formation at the grafted junction [24,45]. WOX4 regulates the vascular cambium development [46]. CDKB1;2 may promote cell cycle progression [47]. TMO6 regulates the vascular and cell-wall-related gene expression at the graft junction [31]. VND7 promotes xylem vessel cell differentiation [32,33]. To investigate whether the exogenous MT treatment affected the expression of genes related to graft union healing, we analyzed the expression characteristics of *CmWIND1* and *CmoWIND1*, *CmWOX4* and *CmoWOX4*, *CmCDKB1;2* and *CmoCDKB1;2*, *CmTMO6* and *CmoTMO6*, and *CmVND7* and *CmoVND7* at different days after grafting (Figure 4 and Figure 5). The results indicate that exogenous MT treatment significantly enhanced the expression of *CmWIND1* and *CmoWIND1*, with a notably pronounced effect on the rootstock. Additionally, the expression level of *CmoWOX4* was significantly higher than that of the control at 1 DAG, although it did not significantly affect *CmWOX4*. At 6 DAG, the expression levels of *CmCDKB1;2* and *CmoCDKB1;2* and *CmTMO6* and *CmoTMO6* were significantly higher than those of the control. Furthermore, the expression levels of *CmVND7* and *CmoVND7* were notably higher than those of the control at 8 DAG. So, exogenous MT treatment could accelerate graft union healing of oriental melon scion grafted onto squash rootstock by increasing the expression levels of genes related to graft union healing. In addition, Chen et al. [7] showed that MT was also involved in regulating plant root growth, and the mustard seedlings treated with low MT concentration could promote their root growth. So, we analyzed the grafted seedlings’ quality after the exogenous MT treatment and found that the root length, root surface area, root average diameter, and the total number of root tips of grafted seedlings with exogenous MT treatment were all higher than those of the control (Figure 9). However, the molecular mechanism of applying exogenous MT to regulate the graft union healing process of oriental melon grafted onto squash rootstock remains to be further studied and clarified. This intricate process involves multiple molecular and biological mechanisms, and understanding its detailed molecular regulatory network is crucial for effectively promoting graft union healing. By delving deeper into the role of melatonin in this process, we can anticipate discovering more key genes, signaling pathways, and biochemical reactions involved, thereby gaining a more profound understanding of how to optimize and enhance plant graft union healing.

## 4. Materials and Methods

### 4.1. Plant Materials

In this study, we used the oriental melon ‘T0948-2’ cultivar (*Cucumis melo* var. *makuwa* Makino) and squash ‘ShengZhen No. 1’ cultivar (*C. moschata*) as the scion and rootstock, respectively. The growth conditions of the rootstock and scion seedlings consisted of 25–28 °C during day and 18–20 °C at night, substrate relative humidity of 60–70%, and 9–10 h of sunlight. The melatonin (300 µmol·L^−1^, MT) was sprayed every two days on the leaf surfaces of the oriental melon and squash when the cotyledon was unfolded, and distilled water was sprayed as the control. When the oriental melon scion grew to one-leaf size, the one-cotyledon graft method was used to graft [48]. The stem tissues of grafted seedlings showing the same growth characteristics were collected at 2, 5, 6, 8, and 9 days after grafting (DAG). The collected samples were immediately transferred to liquid nitrogen and stored at −80 °C. Three bio-replicates were prepared for each treatment.

### 4.2. Determination of Lignin Content and Enzyme Activities

We sampled 0.5 g of stem tissues from the graft junction to measure the lignin contents of grafted seedlings [49], and the enzyme activities related to lignin synthesis were also measured. The CAD and 4CL activities were determined according to Takshak et al. [50], and the PAL activity was measured according to Su et al. [51]. According to the enzyme-linked immunosorbent assay (ELISA) kit (Jiangsu Meimian Industrial Co., Ltd., Yancheng, China) manual, the other enzyme activities (HCT, HCALDH, COMT, CCoAOMT) were also measured. Three separate biological experiments were conducted under the same conditions to replicate results.

### 4.3. Histological Section Observation

The 0.3–0.5cm stems above and below the graft junction of grafted seedlings were sampled. The samples were fixed, softened, dehydrated, infiltrated, and embedded in paraffin, according to Ribeiro et al. [52]. Transverse serial sections approximately 10 μm thick were cut and stained with pH 4.4 toluidine blue [53] and mounted using synthetic resin (Permount). Sections were examined using a light microscope (Lecia RM 2245, Nussloch, Germany). The acid fuchsin absorption assay was also conducted to identify the scion and rootstock connection. We randomly selected five plants, respectively, from thirty plants in the MT treatment and control. The 1.0 cm stem segments above and below the graft junction were cross-cut, the rootstock stems were vertically placed in 1% acidic fuchsin solution for 1 h, and then the 2.5 mm stem segments above the graft union were cross-cut to investigate the acid fuchsin absorption. The acid fuchsin absorption was observed and photographed under a confocal laser microscope.

### 4.4. Quantitative Real-Time PCR (qRT-PCR)

We used qRT-PCR to determine the relative expression of *CmWIND1* and *CmoWIND1*, *Cm CDKB1;2* and *Cmo CDKB1;2*, *CmTMO6* and *CmoTMO6*, *CmVND7* and *CmoVND7*, and *CmCADs* (*CmCAD1*, *CmCAD2*, *CmCAD3*, *CmCAD4*) in the stem tissues of oriental melon and squash rootstock. The qRT-PCR was conducted using a Bio-Rad CFX96 PCR machine and Pro Taq HS SYBR Green premix qPCR kit (AG) in a 20 μL reaction system. The components of the system included 2× SYBR Green Pro Taq HS Premix (10 μL), cDNA (2 μL), forward and reverse primers (0.4 μL each), and ddH_2_O (7.2 μL). The thermal cycling protocol was initial denaturation at 95 °C for 2 min, followed by 40 cycles of 10 s at 95 °C and 30 s at 60 °C [54]. Three separate biological experiments were conducted under the same conditions to replicate the results.

### 4.5. Virus-Induced Gene Silencing (VIGS) Assay

The *CmCAD1* gene was amplified by PCR and cloned into the pTRV2 vector. Subsequently, the TRV2-*CmCAD1* silencing vector was constructed, and pTRV1, pTRV2, and TRV2-*CmCAD1* were transformed into the Agrobacterium EHA105 receptor state [55]. The pTRV1, pTRV2, and TRV2-*CmCAD1* were suspended to OD_600_ = 1.0. Upon exposure to the core, the oriental melon scion seedlings were injected with an infectious liquid via a syringe into the cotyledon. TRV1 and TRV2-*CmCAD1* were blended in a 1:1 ratio, and the combination of TRV1 and TRV2 was then injected into the back side of the oriental melon scions’ cotyledons. TRV-*CmCAD1* and TRV-Control oriental melon lines were obtained. After the injection, the oriental melon scion seedlings were cultured in the artificial light incubator, then in darkness for 18 h, followed by normal day–night alternation. The light temperature was 26 °C, and the dark temperature was 18 °C. The graft was conducted during the one-leaf unfolding. In the graft union healing process, we sampled the stem tissues of the oriental melon scion to measure the relative expression of *CmCAD1* and analyzed the reconnection of the scion and rootstock.

### 4.6. Statistical Analysis

The data are presented as the mean ± standard deviation of three replicate samples and were plotted and statistically analyzed by PraphPad Prism8.0.2 software. Differences in samples were assessed at a significance level (*p* ≤ 0.05, *p* ≤ 0.01, *p* ≤ 0.0001) by a one-way ANOVA test.

## 5. Conclusions

The exogenous MT application significantly increased the enzyme activities related to lignin synthesis and lignin content in the oriental melon scion and squash rootstock stems. Moreover, the lignin accumulation induced by the exogenous MT significantly promoted the graft union healing of oriental melon scion grafted onto squash rootstock and improved the qualities of grafted seedlings.

## Figures and Tables

**Figure 1 ijms-25-03690-f001:**
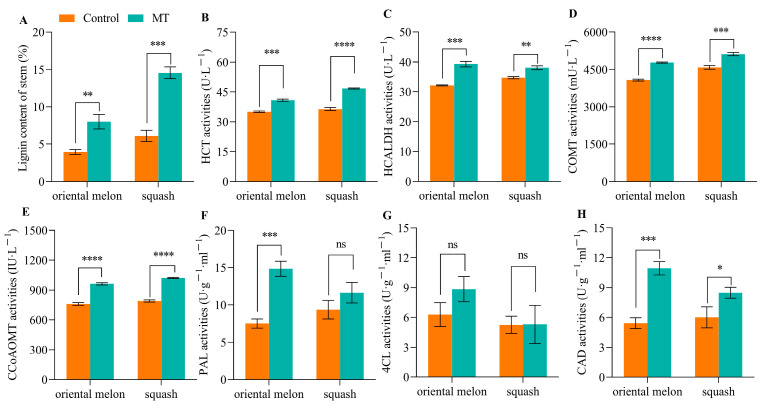
Effects of exogenous MT application on lignin content and enzyme activities related to lignin biosynthesis of oriental melon and squash seedling stems ((**A**), lignin content. (**B**), HCT activities. (**C**), HCALDH activities. (**D**), COMT activities. (**E**), CCoAOMT activities. (**F**), PAL activities. (**G**), 4CL activities. (**H**), CAD activities. MT, melatonin. ns, no significance. *, *p* ≤ 0.05. **, *p* ≤ 0.01. ***, *p* ≤ 0.001. ****, *p* ≤ 0.0001).

**Figure 2 ijms-25-03690-f002:**
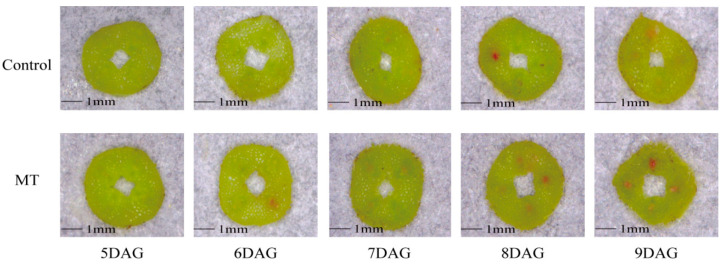
Observation of acid fuchsin absorption during the graft healing process of oriental melon scion grafted onto squash rootstock (MT, melatonin. DAG, days after grafting. Scale bars, 1 mm).

**Figure 3 ijms-25-03690-f003:**
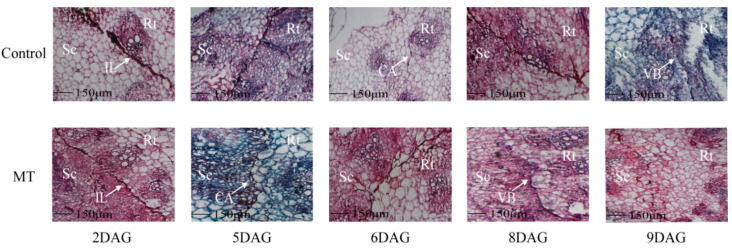
Paraffin section observation during the graft healing process of oriental melon scion grafted onto squash rootstock (IL, the isolated layer. CA, the callus. VB, the vascular bundles. Sc, scion. Rt, rootstock. MT, melatonin. DAG, days after grafting. Scale bars, 150 µm).

**Figure 4 ijms-25-03690-f004:**
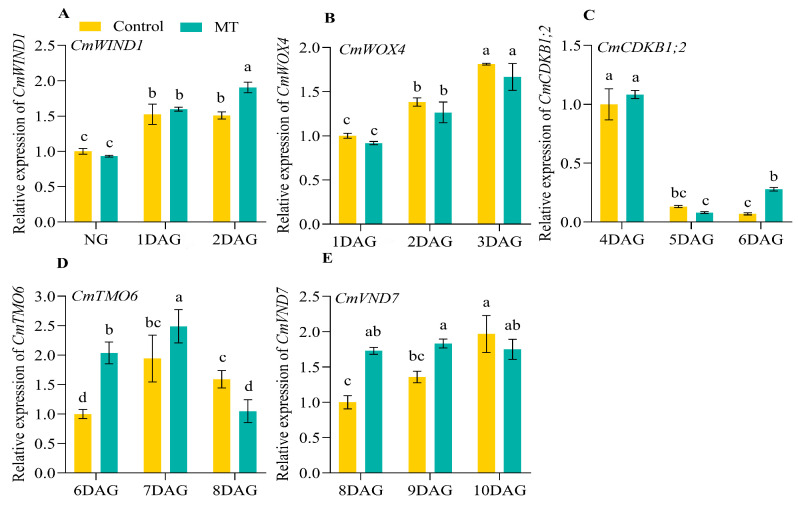
The relative expression of genes related to graft union healing in the oriental melon scion during the graft healing process ((**A**), *CmWIND1*. (**B**), *CmWOX4*. (**C**), *CmCDKB1;2*. (**D**), *CmTMO6*. (**E**), *CmVND7*. MT, melatonin. NG, no grafting. DAG, days after grafting. Different letters indicate significant differences, *p* ≤ 0.05).

**Figure 5 ijms-25-03690-f005:**
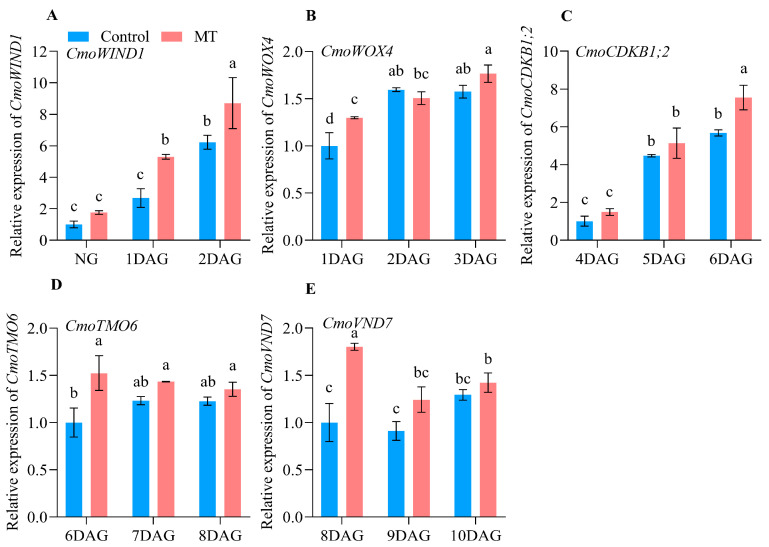
The relative expression of genes related to graft union healing in the squash rootstock during the graft healing process ((**A**), *CmoWIND1*. (**B**), *CmoWOX4*. (**C**), *CmoCCDKB1;2*. (**D**), *CmoTMO6*. (**E**), *CmoVND7*. MT, melatonin. NG, no grafting. DAG, days after grafting. Different letters indicate significant differences, *p* ≤ 0.05).

**Figure 6 ijms-25-03690-f006:**
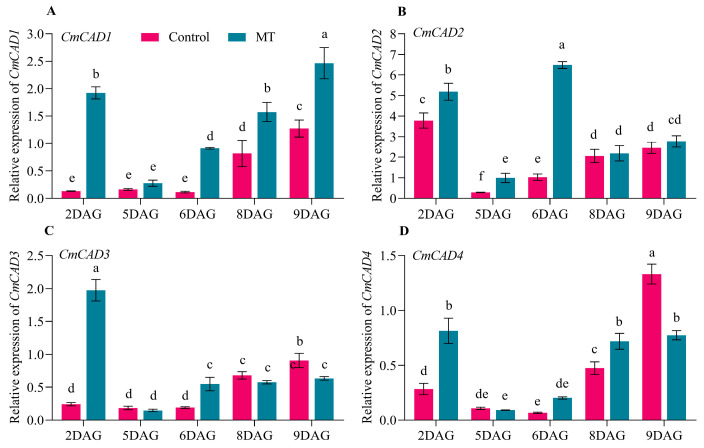
The relative expression of *CmCADs* in the oriental melon scion during the graft healing process ((**A**), *CmCAD1*. (**B**), *CmCAD2*. (**C**), *CmCAD3*. (**D**), *CmCAD4*. MT, melatonin. DAG, days after grafting. Different letters indicate significant differences, *p* ≤ 0.05).

**Figure 7 ijms-25-03690-f007:**
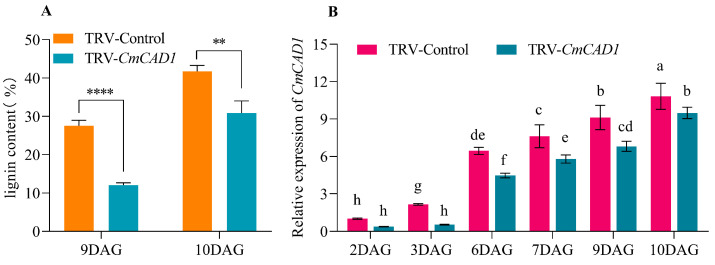
The relative expression of *CmCAD1* and lignin content of oriental melon scion stem after the transiently silenced *CmCAD1* during the graft healing process ((**A**), lignin content. (**B**), the relative expression levels of *CmCAD1*. DAG, days after grafting. Different letters indicate significant differences, **, *p* ≤ 0.01. ****, *p* ≤ 0.0001).

**Figure 8 ijms-25-03690-f008:**
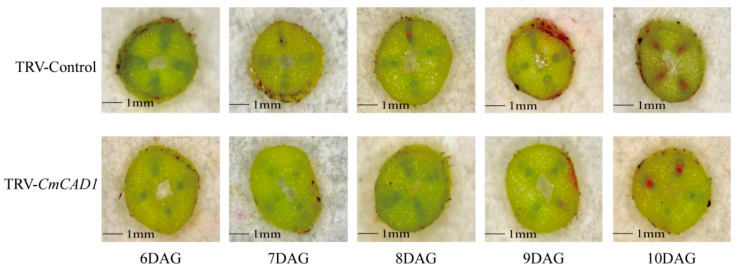
Observation of acid fuchsin absorption after the transiently silenced *CmCAD1* during the graft union healing process of oriental melon scion grafted onto squash rootstock (DAG, days after grafting. Scale bars, 1 mm).

**Figure 9 ijms-25-03690-f009:**
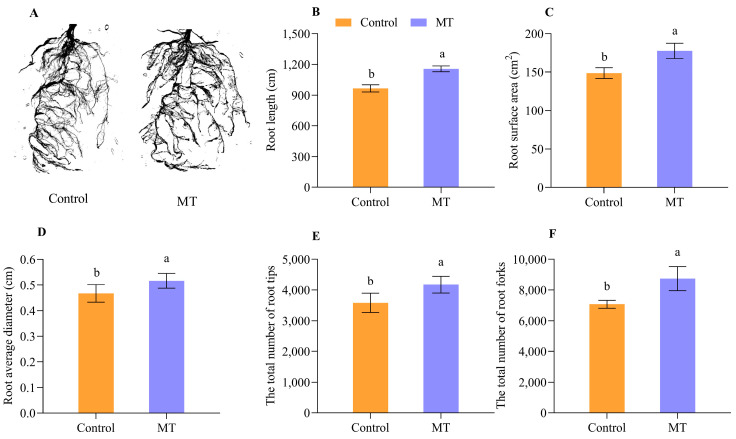
Analysis of root growth characteristics of five-leaf grafted seedlings treated with the exogenous MT ((**A**), photographs of roots. (**B**), root length. (**C**), root surface area. (**D**), root average diameter. (**E**), the total number of root tips. (**F**), the total number of root forks. MT, melatonin. Different letters indicate significant differences, *p* ≤ 0.05).

## Data Availability

The data presented in this study are available on request from the corresponding author.

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
