# Peer review of "Exogenous Melatonin Application Accelerated the Healing Process of Oriental Melon Grafted onto Squash by Promoting Lignin Accumulation"

_ijms, 2024, doi:10.3390/ijms25073690_

Round 1

Reviewer 1 Report

Comments and Suggestions for Authors

Dear Authors

The type of research seems interesting, although it leans more towards fantasy than practicality.

In the text, the items that should be corrected by the authors are mentioned as comments, especially in the materials and methods section.Please don't forget to answer the following two comments:

Do you think greenhouse conditions (such as temperature, humidity,…) are more important or the use of substances such as melatonin during the grafting process?

Would you recommend using melatonin during grafting in normal conditions or only in abnormal conditions (such as stress)?

Author Response

Dear Reviewer,

Thanks for your generous comments about the article. We had made some revisions according to your comments. Please check it in the Coverletter. Thank you again for your assistance.

Sincerely yours,

Chuanqiang Xu (C.X.)

Reviewer 2 Report

Comments and Suggestions for Authors

The authors use qRT-PCR and enzymatic assays to demonstrate convincingly that melatonin induces the activity of several enzymes, especially within the phenylpropanoid biosynthetic pathway.  Anywhere that the data are quantified and analyzed using standard statistical tools (Figures 1,4,5,6,7,9), the results are convincing (and have reportedly been replicated three times).  

More questionable are the acid fuschin transport assays (Figure 1, 8).  The Methods section says that three "bio-replicates" were analyzed.  Does that mean only three plants at each treatment?  This seem like it is too few.  More importantly, how many plants out of the total number of plants assays in each treatment had the pattern shown in the photographs?  Ideally, there would be more than three plants tested at each treatment, and each panel of the figure would show how many plants showed the pattern presented (e.g. 8/10; 9/10 ?).  This is even more important in the VIGS treatments, given the added variability due to the silencing process.  In each case, some quantitative information is necessary to trust the conclusion that MT is improving graft healing.

The manuscript is otherwise very well-written and well-organized. I have the following additional minor suggestions.

1. Include a brief explanation of the reason for studying this process in the context of oriental melon/squash.  Is there an economically relevant application of grafting these species, or are there particular scientific justifications for using these species to study this process.

2. Italicize the latin names in line 47 and else where

3. Correct the spelling of "Histoligucal section observation" in the methods section.

Author Response

(The authors gave the same response as above.)
